# PREDICTING OBSERVATION AFTER ACTION IN A HIERARCHICAL ENERGY-BASED MODEL WITH MEMORY

## ABSTRACT

Understanding the mechanisms of brain function is greatly advanced by predictive models. Recent advancements in machine learning further underscore the potency of prediction for learning optimal representation. However, there remains a gap in creating a biologically plausible model that explains how the neural system achieves prediction. In this paper, we introduce a framework employing an energy-based model (EBM) to capture the nuanced processes of predicting observation after action within the neural system, encompassing prediction, learning, and inference. We implement the EBM with a hierarchical structure and integrate a continuous attractor neural network for memory, constructing a biologically plausible model. In experimental evaluations, our model demonstrates efficacy across diverse scenarios. The range of actions includes eye movement, motion in environments, head turning, and static observation while the environment changes. Our model not only makes accurate predictions for environments it was trained on, but also provides reasonable predictions for unseen environments, matching the performances of machine learning methods in multiple tasks. We hope that this study contributes to a deep understanding of how the neural system performs prediction.

## 1 INTRODUCTION

To survive, humans need to interact with environment through actions, requiring an understanding of how these actions impact the surroundings. This involves building an internal model in the brain to represent the outside world Knill & Pouget (2004); Friston & Price (2001). The success of large language models (LLMs) in understanding token-based worlds also indicates that predicting the next observation is a good objective in learning representations Radford et al. (2017). However, as humans living in the physical world, our received observations are high-dimensional and diverse. This presents a challenge in understanding how the brain predicts the next observation.

The world model Schmidhuber (1990); LeCun (2022) has laid out a basic framework for prediction. Recently, the machine learning society has made significant progress in predicting high-dimensional observations through planning in latent spaces Ha & Schmidhuber (2018); Hafner et al. (2019a; 2023); Nguyen et al. (2021). However, these models, not designed for explaining the neural system, lack consideration for biological realism Chung et al. (2015), and they typically employ biologically implausible training algorithms such as backpropagation (BP) or backpropagation through time (BPTT). In the neuroscience society, there are ongoing efforts to model the hippocampal-entorhinal system as sequential generative models Whittington et al. (2018); George et al. (2023). However, these approaches either employed a variational method, leading to still requiring BPTT, or assumed access to the underlying state of the world, which is not realistic to the neural system.

Energy-based models (EBMs) Ackley et al. (1985) provide a framework for inference with sampling methods and learning with Hebb's rule. The variability of neuronal responses in the brain has been explained as Monte Carlo sampling Hoyer & Hyvärinen (2002), which naturally accounts for the regular firing and other response properties of biological neurons Haefner et al. (2016); Orbán et al. (2016); Echeveste et al. (2020). Hebb's rule is a widely observed local learning rule in the neural system. A recent work Dong & Wu (2023) has shown that hierarchical EBMs are capable of learning complex probability distributions, suggesting their potential widespread applications in the brain.

In this paper, we propose a sequential generative model based on hierarchical EBMs to capture how the brain predicts the next observation after an action (Section 3). In our model (Section 4), a Markov chain of latent variables is employed, whose conditional probabilities following Gaussian distributions. This choice helps us bypass the computation of the partition function, leading to accelerated model convergence. Furthermore, the introduction of error neurons ensures that the learning process is localized. We also utilized a continuous attractor neural network (CANN) Amari (1977); Ben-Yishai et al. (1995); Wu et al. (2008) to memorize past events to improve prediction. In the brain, sensory information undergoes hierarchical processing through the cortex before entering higher brain regions (such as the IT region and hippocampus) DiCarlo et al. (2012), while CANNs have been widely used as canonical models for elucidating the memory process in these higher brain regions Wills et al. (2005). In the experiments (Section 5), we considered visual observations and constructed various actions to assess model performances. The actions include eye movement, motion in a virtual environment, motion and head-turning in a real environment, and static observation while the external world varies. Our model demonstrates effective predictions for the environments it was trained on, and the model also generates reasonable predictions for unseen environments. In several tasks, our biologically plausible model has achieved performances on par with machine learning methods. Key contributions of this work are summarized as follows:

- **Energy-based Recurrent State Space Model (RSSM)** We introduce a novel framework for RSSM grounded in energy-based principles. This approach diverges from the conventional variational RSSM Chung et al. (2015); Hafner et al. (2019b) by offering distinct methodologies for inference, learning, and prediction within the energy-based paradigm.

- **Biologically Inspired RSSM Implementation** Our implementation leverages hierarchical EBMs and CANNs to realize the RSSM. The learning mechanism is characterized by its local properties, both spatially and temporally, without relying on BP or BPTT. Algorithms for inference and prediction can be implemented through neural dynamics.

- **Establishment of a Prediction Error Upper Bound** Setting our approach apart from previous methodologies that employ free energy or the evidence lower bound (ELBO) as the loss function, we adopt the prediction probability distribution within the latent space for sampling. This provides a novel perspective on model optimization.

## 2 RELATED WORK

**The world model** Schmidhuber (1990); LeCun (2022) laid out a framework for predicting observations following an agent's action. Recently, RSSM compresses observations through a variational autoencoder (VAE) , then performs predictions in the compressed temporal space using temporal prediction models like RNNs Chung et al. (2015); Hafner et al. (2019b), Transformers Chen et al. (2022), S4 models Samsami et al. (2024) or continuous Hopfield networks Whittington et al. (2018). Our model adopts this RSSM framework but innovates by incorporating EBMs instead of VAEs and utilizing CANNs for temporal predictions. Our model aligns with biological plausibility, departing from less biologically realistic architectures and training methods.

**Active inference** Friston et al. (2017); Smith et al. (2022) is another framework which can predict the observation after an action. These works are unified under the free energy principle framework Friston (2010), modeling the prediction process as a hidden Markov model (HMM, a special case of the RSSM), and using the variational message passing algorithm for inference Da Costa et al. (2020); Parr et al. (2019). We model the entire process as an RSSM with a temporal model (CANN) that can compress all previous states rather than reliance on the current state alone. Also our model employs a sampling algorithm, enabling online inference and learning without the need to know the entire sequence.

**Predictive coding networks (PCNs)** Rao & Ballard (1999) can be viewed as an implementation of EBMs, with most current PCN works deal with static inputs Salvatori et al. (2021; 2023); Millidge et al. (2022), do not involve temporal prediction of the next observation after actions. A recent study, ActPC Ororbia & Mali (2023), does introduce actions within the Markov process (a special case of HMM); however, it lacks an encoder-decoder structure, assuming an identity matrix mapping between observations and latent states. Our model integrates an EBM as the encoder-decoder part. Furthermore, while their approach utilizes a buffer to store observations directly, our model employs a CANN to efficiently compress all previous states.

## 3 ENERGY-BASED RECURRENT STATE SPACE MODEL

To build an internal model capturing the change of the environment during interaction, the brain needs to learn to predict the next observation after an action. We consider that the brain employs an energy-based RSSM as the intrinsic generative model for generating predictions. This section outlines the model framework encompassing the predicting, learning, and inference processes, with the detailed neural implementation presented in Section 4.

**Problem setup.** Consider an action $a_t$ taken at time $t$. The brain anticipates the observation $o_t$ the sensory neurons will receive after this action. According to the laws of physics, the world is Markovian, i.e., the next moment is solely determined by the previous moment. However, our observation $o_t$ and action $a_t$ reflect only a subset of the world, and we do not have the full knowledge of the world. To enhance the prediction, the brain can rely on past experiences. Denote $K_t = \{o_{<t}, a_{\le t}\}$ the observation-action sequence we have experienced, upon which the brain can build a memory trace $m_t$ to facilitate the next prediction of $o_t$.

**Generative model.** We consider that the brain utilizes the marginal distribution of the intrinsic generative model (Figure 1a) to approximate the distribution of the true observation. The joint distribution at time $t$ is expressed as,

$$p_\theta(o_t, s_t|m_t) = p_\theta(o_t|s_t)p(s_t|m_t), \tag{1}$$

where $s_t$ are latent variables represented by neuronal responses. We employ the sampling-based probabilistic representation Hennequin et al. (2014); Dong et al. (2022), assuming that the neural activity at time $t$ is a sample of the random variable $s_t$. $p_\theta(o_t|s_t)$ is the likelihood function and $p(s_t|m_t)$ can be regarded as the prior of the latent variable before receiving the observation given the memory state $m_t$. At the next time step $t+1$, the memory is updated following a transition probability $m_{t+1} \sim p(m_{t+1}|m_t, s_t, a_{t+1})$, which contains the information of the past experiences $K_{t+1}$. In this paper, we take this transition probability as a Dirac delta function, which makes our generative model essentially a recurrent state-space model Hafner et al. (2019b).

**Prediction.** To predict the upcoming observation after the action $a_t$, the brain needs to generate samples following the marginal distribution $p_\theta(o_t|m_t)$. According to the generative model in Eq.(1), the brain first generates the latent variable $\hat{s}_t \sim p(s_t|m_t)$ and then the observation $\hat{o}_t \sim p_\theta(o_t|\hat{s}_t)$ (Figure 1b).

**Learning.** After receiving the true observation $o_t \sim p_{\text{true}}(o_t)$, the brain will update the generative model to improve future prediction. The disparity between the prediction and the true observation can be quantified by the cross-entropy $\mathcal{H}$, expressed as,

$$\mathcal{H} = -\mathbb{E}_{o_t \sim p_{\text{true}}(o_t)} \log p_\theta(o_t|m_t), \tag{2}$$

$$\le \underbrace{-\mathbb{E}_{o_t \sim p_{\text{true}}(o_t)} \mathbb{E}_{\hat{s}_t \sim p(s_t|m_t)} \log p_\theta(o_t|\hat{s}_t)}_{=:\mathcal{L}}. \tag{3}$$

We use an energy-based model with parameters $\theta$ to model the likelihood function,

$$p_\theta(o_t|s_t) = \frac{\exp\left[-E_\theta(o_t, s_t)\right]}{Z_\theta}, \quad Z_\theta = \int \exp\left[-E_\theta(o_t, s_t)\right] \mathrm{d}o_t. \tag{4}$$

where $E_\theta(o_t, s_t)$ is the energy and $Z_\theta$ is the partition function, Since calculating the cross-entropy in Eq.(2) involves complicated integration, which makes it intractable, we choose its upper-bound $\mathcal{L}$ defined in Eq.(3) as our learning objective. Equivalently, -$\mathcal{L}$ can be interpreted as the lower bound of the mutual information between the neural prediction and the observation (see deviation in Appendix A). The neural system can adopt a gradient-based learning method such as gradient decent, and the gradient of $\mathcal{L}$ is calculated to be (Figure 1b dashed lines),

$$\nabla_\theta \mathcal{L} = \mathbb{E}_{\hat{s}_t \sim p(s_t|m_t)} \left[ \mathbb{E}_{o_t \sim p_{\text{true}}(o_t)} \nabla_\theta E_\theta(o_t, \hat{s}_t) - \mathbb{E}_{\hat{o}_t \sim p(o_t|\hat{s}_t)} \nabla_\theta E_\theta(\hat{o}_t, \hat{s}_t) \right]. \tag{5}$$

**Inference & memory update.** After updating the likelihood function $p_\theta(o_t|s_t)$, the brain also needs to update the neural representation $s_t$ and the memory representation $m_t$. Specifically, the new distribution of $s_t$ becomes,

$$p_{\text{post}} = \arg\max_q \mathbb{E}_q \log p_\theta(o_t|s_t) - D_{\text{KL}}\left[q||p(s_t|m_t)\right] \propto p_\theta(o_t|s_t)p(s_t|m_t). \tag{6}$$

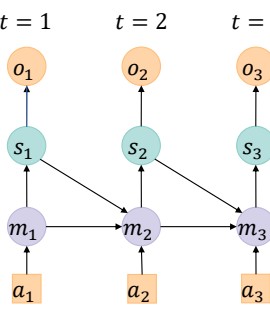
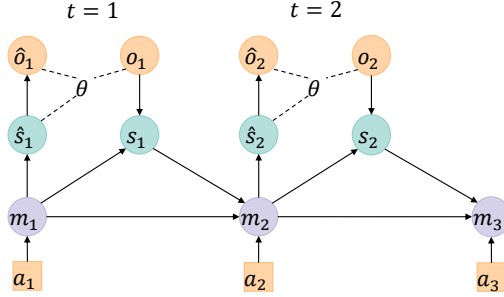

(a) Generative model          (b) Prediction learning & inference

Figure 1: (a) The directed graphical model of the generative model. Taking $t = 2$ as an example, $s_2$ follows the prior, $o_2$ follows the likelihood function, and $m_3$ follows the transition probability. (b) Taking $t = 2$ as an example, the brain initially generates a prediction $\hat{s}_2$ for the neural activity, followed by producing a prediction $\hat{o}_2$ for the observation. Then the network parameters are updated, as indicated by the dashed lines, and the posterior for the current time step is obtained. At last, the memory is updated based on action $a_3$ and the sample $s_2$ following the posterior.

---

**Algorithm 1:** General process

**while** *still alive* **do**
    // from time $t$ to $t + 1$
    Sample $\hat{s}_t \sim p(s_t|m_t)$, $\hat{o}_t \sim p_\theta(o_t|\hat{s}_t)$;
    Receive $o_t \sim p_{\text{true}}(o_t)$;
    Update $\theta$ by minimize $\mathcal{L}$ using $\nabla_\theta \mathcal{L}$;
    Sample $s_t \sim p_{\text{post}}(s_t)$ ;
    Update $m_{t+1} \sim p(m_{t+1}|m_t, s_t, a_{t+1})$.

---

This new distribution reflects that, on one hand, under this distribution, we can better predict the true observation, i.e., maximizing $\mathbb{E}_q \log p_\theta(o_t|s_t)$. This target also implies enabling $s_t$ to contain as much information from $o_t$ as possible (refer to the deviation in Appendix B). Meanwhile, we aim to minimize variation in the neural representation, i.e., ensuring that the new distribution remains close to the previous. To strike a balance between these two objectives, the new distribution takes the form of the $p_{\text{post}}$.

The brain can use the sampling-based approach to obtain the distribution of $p_{\text{post}}$, such as the Langevin dynamic,

$$\tau_s \frac{\mathrm{d}s}{\mathrm{d}t} = \nabla_s \log p_\theta(o_t|s_t) + \nabla_s \log p(s_t|m_t) + \sqrt{2\tau_s}\xi, \tag{7}$$

where $\xi$ is Gaussian white noise and $\tau_s$ is the time constant. At last, we use the samples of the distribution $p_{\text{post}}(s_t)$ to update the memory according to the generative model,

$$m_{t+1} \sim p(m_{t+1}|m_t, s_t, a_{t+1}), \quad s_t \sim p_{\text{post}}(s_t). \tag{8}$$

Algorithm 1 outlines the general procedure by which the neural system continually engages in prediction, learning and memory updating.

## 4 A HIERARCHICAL NEURAL NETWORK MODEL

In this section, we propose a hierarchical neural network to implement the above generative model, and outline the specific dynamics involved in prediction, learning, and inference, as discussed in Section 3. Approximating the target distribution $p_{\text{true}}(o_t)$, which is diverse and complex, requires a good representation ability of the model. The hierarchical structure has been demonstrated to have strong expressive power and is widely adopted in the biological neural systems. Moreover, we employs a continuous attractor neural network (CANN) to model the memory process. All vectors below are column vectors, and all multiplications are matrix multiplications.

**A hierarchical generative model.** Let $s_t^0 \in \mathbb{R}^{n_0}$ be the observation variable. There are $L$ layers of neurons representing the latent variables $s_t^{1:L} = \{s_t^1, s_t^2, ..., s_t^L\}$, $s_t^l \in \mathbb{R}^{n_l}$. The joint distribution is a Markov chain,

$$p_\theta(s_t^{0:L}|m_t) := p(s_t^L|m_t) \prod_{l=0}^{L-1} p_\theta(s_t^l|s_t^{l+1}). \tag{9}$$

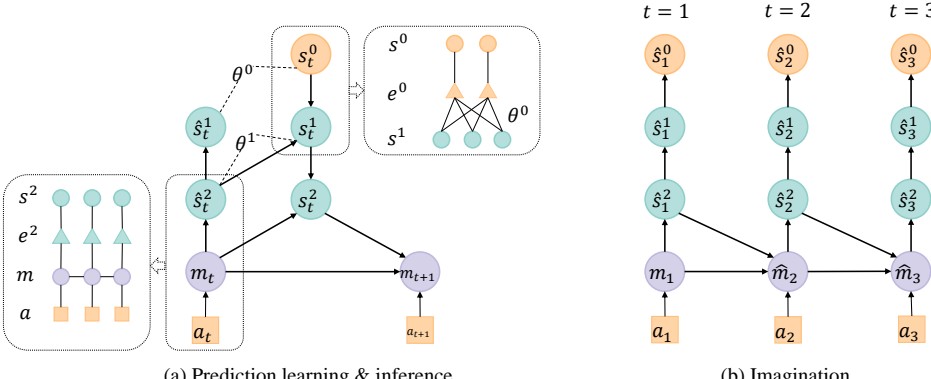

(a) Prediction learning & inference    (b) Imagination

Figure 2: (a) In the hierarchical structure, the activity of neurons in the upper layer is the observation for the neurons in the lower layer. In the case of $L = 2$, $s_t^0$ is the observation of $s_t^1$, and $s_t^1$ is the observation of $s_t^2$. The bottom-left box illustrates the connection of the memory-representing CANN with action neurons and its link to $s^2$ through $e^2$. The top-right box shows the connections between $s^0$ and $s^1$ through $e^0$. (b) Based on the current memory $m_1$, we imagine the observations we would receive at each time step after taking a series of actions.

To avoid calculating the deviation of the partition function in the likelihood function, we utilize the Gaussian distribution, whose partition function is constant,

$$p_\theta(s_t^l | s_t^{l+1}) := \mathcal{N}(s_t^l;\ \theta^l f(s_t^{l+1}),\ (\Lambda^l)^{-1}), \tag{10}$$

where $f(\cdot)$ is the element-wise activation function and $\Lambda^l$ is the inverse of the covariance matrix called precision matrix. $\theta_l \in \mathbb{R}^{n_l \times n_{l+1}}$ are parameters which determine the connectivity structure of the network.

**The memory network.** Attractor neural networks have been widely used as canonical models for elucidating the memory process in the neural system Amari (1972); Hopfield (1982). Among these, CANNs excel in capturing continuous variables, as the underlying state of a temporal sequence is typically continuous. Therefore, we use the activity of CANN neurons at time $t$ as the memory $m_t$. The CANN, through its recurrent connections, forms a series of continuous attractors. This sequence of attractors constitutes a stable low-dimensional manifold, serving as the memory space. In spatially-related tasks, it is also referred to as a cognitive map O'keefe & Nadel (1979); Samsonovich & McNaughton (1997). The CANN receives inputs from actions and from the last layer's latent neurons for prediction and memory update (see Appendix C for the CANN dynamics). When the CANN receives inputs from action $a_t$ and neurons $s_{t-1}^L$, it generates activity $m_t$, which gives rise the neuronal activity in the last layer according to,

$$p(s_t^L | m_t) := \mathcal{N}(s_t^L;\ m_t,\ (\Lambda^L)^{-1}). \tag{11}$$

**Prediction.** At time $t$, the neural network first generates the prediction samples $\hat{s}_t^{0:L}$ from layer $L$ to 0 according to,

$$\hat{s}_t^L \sim p(s_t^L | m_t), \quad \hat{s}_t^l \sim p(s_t^l | \hat{s}_t^{l+1}). \tag{12}$$

We use the Langevin dynamic to generate predictions,

$$\tau_s \frac{ds^l}{dt} = \nabla_{s^l} p(s^l | \hat{s}_t^{l+1}) + \sqrt{2\tau_s}\xi = -\Lambda^l \hat{e}_t^l + \sqrt{2\tau_s}\xi, \tag{13}$$

where $\hat{e}_t^l = s_t^l - \theta^l f(\hat{s}_t^{l+1})$ and $\hat{e}_t^L = s_t^L - m_t$ are the value represented by error neurons. We adopt the idea of predictive coding networks Rao & Ballard (1999); Whittington & Bogacz (2017) by introducing error neurons, to satisfy Hebb's rule during learning. The connectivity diagram of neurons is depicted by the dashed box in Figure 2a.

**Learning & inference.** After the model receives the observation $s_t^0 \sim p_{\text{true}}(s_t^0)$, for $s^1$ represented by neurons in layer one, the likelihood function is $p(s_t^0 | s_t^1)$ and the prior distribution is $p(s_t^1 | \hat{s}_t^2)$. Thus, the prediction bound $\mathcal{L}_t^0$ can be written as,

$$\mathcal{L}_t^0 := -\mathbb{E}_{s^0 \sim p_{\text{true}}(s_t^0)} \mathbb{E}_{s^1 \sim p(s_t^1 | \hat{s}_t^2)} \log p(s_t^0 | s_t^1) = \frac{1}{2}\left(\hat{e}_t^0\right)^T \Lambda^0 \hat{e}_t^0 + C^0, \tag{14}$$

where $C^0$ is the constant. Then, synaptic parameters $\theta^0$ are updated to minimize $\mathcal{L}_t^0$ using gradient descent,

$$\tau_\theta \frac{d\theta^0}{dt} = -\nabla_{\theta^0} \mathcal{L}_t^0 = \Lambda^0 \hat{e}_t^0 f(\hat{s}_t^1)^T. \tag{15}$$

This shows that the synaptic changes are determined solely by local neurons, adhering to Hebb's rule. Then neurons in layer one keeps a balance between the likelihood and the prior by inferring the posterior $p_{\text{post}}^1 \propto p(s_t^0|s_t^1)p(s_t^1|\hat{s}_t^2)$ through Langevin dynamic,

$$\tau_s \frac{ds_t^1}{dt} = \nabla_{s_t^1} \log p(s_t^0|s_t^1) + \nabla_{s_t^1} p(s_t^1|\hat{s}_t^2) + \sqrt{2\tau_s}\xi = f'(s_t^1) \odot (\theta^0)^T \Lambda^0 e_t^0 - \Lambda^1 \hat{e}_t^1 + \sqrt{2\tau_s}\xi, \tag{16}$$

where $e_t^0 = s_t^0 - \theta^0 f(s_t^1)$ and $\odot$ is the element-wise product. Then the sample $s_t^1$ following the posterior will be used to minimize $\mathcal{L}_t^1$ and obtain sample $s_t^2 \sim p_{\text{post}}^2$. Each layer will repeat this process and propagate information downward until the last layer (see second for-loop in Algorithm 2). For random vector $s^l$ in layer $l$. The prediction bound $\mathcal{L}_t^l$ is calculated as,

$$\mathcal{L}_t^l := -\mathbb{E}_{s_t^{l-1} \sim p_{\text{post}}^{l-1}} \mathbb{E}_{s_t^l \sim p(s_t^l|\hat{s}_t^{l+1})} \log p(s_t^{l-1}|s_t^l) = \frac{1}{2}\left(\hat{e}_t^l\right)^T \Lambda^l \hat{e}_t^l + C^l, \tag{17}$$

where $C^l$ is the constant. The posterior $p_{\text{post}}^l$ of variables in layer $l$ is calculated as,

$$p_{\text{post}}^l \propto p(s_t^{l-1}|s_t^l)p(s_t^l|\hat{s}_t^{l+1}). \tag{18}$$

---

**Algorithm 2:** Hierarchical neural process

---

**while** *still alive* **do**

  // from time $t$ to $t+1$

  Sample $\hat{s}_t^L \sim p(s_t^L|m_t)$;

  **for** $l \leftarrow L-1$ **to** $0$ **do**

    Sample $\hat{s}_t^l \sim p(s_t^l|\hat{s}_t^{l+1})$;

  Receive observation $s_t^0$;

  **for** $l \leftarrow 0$ **to** $L-1$ **do**

    Update $\theta^l$ by $\frac{d\theta^l}{dt} = -\nabla_{\theta^l} \mathcal{L}_t^l$;

    Sample $s_t^{l+1} \sim p_{\text{post}}^{l+1}$ by

    $\tau_s \frac{ds_t^{l+1}}{dt} = \nabla_{s_t^{l+1}} \log p_{\text{post}}^{l+1} + \sqrt{2\tau_s}\xi$

    ;

  Update $m_{t+1} \sim p(m_{t+1}|m_t, s_t^L, a_{t+1})$.

---

After the variables in layer $L$ converge to their posterior, they serve as inputs to the CANN. Meanwhile, the action $a_{t+1}$ at time $t+1$ is also fed into the CANN. The CANN, following its dynamics, reaches a new steady state with the neuron activity denoted as $m_{t+1}$. Subsequently, the brain utilizes $m_{t+1}$ as the memory to initiate a new round of prediction. Algorithm 2 illustrates the entire process of neural implementation.

**Imagination.** After our model learns to predict the next observation, it acquires an intrinsic representation of the dynamics of the external environment. If we want to know the outcome of a certain action, there is no need to actually perform the action; instead, we can rely on the model to predict the observation we would receive, called imagination. We can continually make predictions in the latent space, incorporating them into memory, and forecast observations after a sequence of actions (Figure 2b).

## 5 EXPERIMENT

We evaluate our model by selecting four types of action in different environments, including eye movement, motion in a virtual environment, motion and head-turning in a real environment, and static observation while the external world varies. To simulate the high-dimensional inputs received by the brain, all observations in our study are exclusively chosen to be visual inputs. We refer to the appendix for hyper parameters (Appendix E).

**Eye movement** refers to changing the direction of the eyeballs to obtain different visual inputs. It stands as the most frequent actions performed by humans, helping us gather as much visual information as possible. To avoid dizziness caused by rapid eye movement, neurons in the posterior parietal cortex encode stimuli that will be seen after planned eye movements Cui & Andersen (2011); Kuang et al. (2016). Additionally, experiments Seung (1996) suggest that neurons in the medial vestibular nucleus form a CANN to record eye direction.

We utilized the CIFAR-10 and Fashion-MNIST datasets to simulate the environments observed by the model. Each image in the dataset is divided into $4 \times 4$ patches, with each patch serving as an

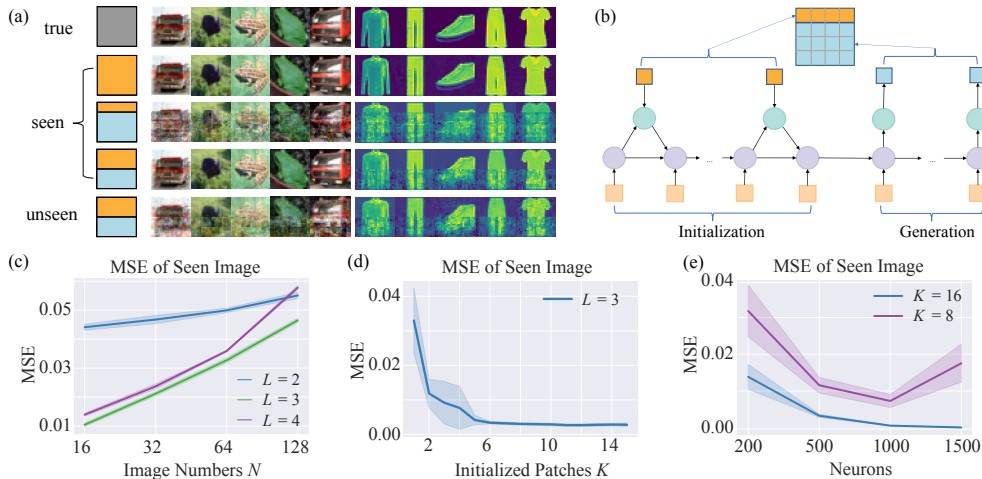

Figure 3: Experiments on modeling eye movement. (a) Generation results of the entire image through initialized memory. The orange and blue areas in the first column indicate the patches used for initialization and those that need to be predicted. 'Seen' refers to images that were used during training, while 'unseen' refers to images that were not used during training (here, both are the same images because different models were used). (b) Testing phase: the first is step is initialization, similar to the training process, involves executing actions, observing the environment, inferring the latent state, and finally updating the memory. However, unlike training, network weights are not updated in this step. The second step involves executing random actions and predicting observations (c) Model tested by memory initialized with $K = 4$ patches. (d) Model trained on $N = 32$ images. When initialized with $K = 6$ patches , the whole image can be almost completely reconstructed. (e) The $x$-axis neuron numbers represent the number of neurons in each layer, with a total of $L = 3$ layers in the network.

input for a single observation to mimic the human receptive field. After each eye movement, the next observation becomes the corresponding patch.

During the learning phase (Figure 2a), we randomly generate a sequence of eye movement, and change the complete image every once in a while. We employed $N = 16, 32, 64, 128$ images for each model, resulting in a total of $N_{\text{tol}} = 16 \times N$ possible observations. Rows 2-4 of Figure 3a demonstrate the generation results for images encountered during training, while the 5th row illustrates the generation results for unseen images. Figure 3c shows the mean squared error (MSE) between the prediction and the ground truth for different network structures (total number of neurons is the same, $L$ varies), which decreases with training epochs. To increase training efficiency, we used a batch size of 128, and roughly, the effectiveness of one epoch can be considered as the average over 128 time steps. Figure 3d depicts the decreases of loss $\mathcal{L}_t^l$ for each layer in the $L = 3$ model across training epochs. Here, the phenomenon of gradient vanishing is observed, and we plan to address this by introducing skip connections to deepen the network. Figure 3e displays the impact of network capacity on performance. When initializing memory with $K = 16$ patches, a higher number of neurons corresponds to improved model performance. However, when initializing with $K = 8$ patches, an excessive number of neurons increases the initialization space, posing a challenge and resulting in a decline in model performance.

| Images $N$ | 16 | | 32 | | 64 | | 128 | |
|---|---|---|---|---|---|---|---|---|
| Patches $K$ | Ours | TDM | Ours | TDM | Ours | TDM | Ours | TDM |
| 4 | 0.0907 | 0.1678 | 0.0834 | 0.1431 | 0.0802 | 0.1281 | 0.0770 | 0.1179 |
| 8 | 0.0687 | 0.1321 | 0.0629 | 0.1272 | 0.0612 | 0.1130 | 0.0606 | 0.0911 |
| 16 | 0.0388 | 0.0532 | 0.0336 | 0.0512 | 0.0304 | 0.0482 | 0.0287 | 0.0471 |

Table 1: MSE of predictions calculated on modeling eye movement. We compare our model with transdreamer Chen et al. (2022) for different Image numbers and initialized patches.

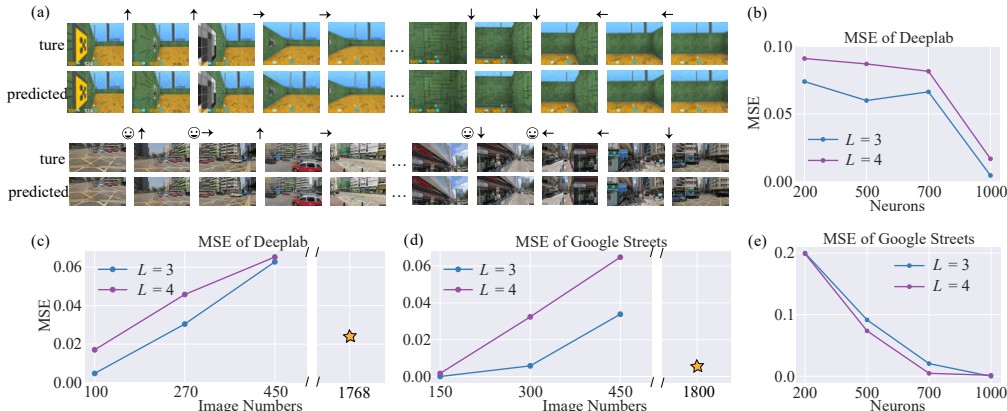

Figure 4: Experiments on modeling motion and head turning. (a) The prediction results. Arrow labels indicate the direction of movement actions, while the combination of the head and arrow illustrates the direction of head-turning actions. (b)(e) The $x$-axis neuron numbers represent the number of neurons in each layer, with training using 150 images. (c)(d) In both figures, the total number of neurons in the model is 3k, and orange stars mark the results when the number of neurons is 10k.

During the testing phase (Figure 3b), we start by randomly initializing the memory. We select $K$ patches from an image and perform random eye movement on these patches for prediction and inference without altering the network weights. After $2K$ steps, the obtained memory becomes the initialized memory. We then use this memory to envision each patch, generating the whole image. Although the CIFAR-10 dataset has higher dimensions than Fashion-MNIST, we found that the model performs better on CIFAR-10. This may be attributed to the fact that CIFAR-10, as a natural image dataset, exhibits higher correlations between patches, which is more favorable for the model's predictive capabilities. We also compared our model's test results on CIFAR-10 with the commonly used TransDreamer model (TDM) Chen et al. (2022) for similar tasks in machine learning methods. The results show that our model achieved better performance with the same number of parameters (Table 1). In appendix D, we provide more details about the training process and clearer generated results.

**Motion and head-turning** alter our spatial position and the orientation of our head, respectively. Experimental findings reveal neurons encoding position and head orientation in the brain exhibit structures akin to CANNs Wills et al. (2005); Kim et al. (2017). Notably, place cells, particularly located in the hippocampus, are believed to be closely associated with spatial cognition and memory functions Moser et al. (2015).

In our experiments, we employed an agent moving in four directions within the Deeplab map Beattie et al. (2016), constructing a dataset for a virtual environment using observed images and action sequences. Additionally, we utilized the Google Street View Static API to capture images by continuously moving and rotating the viewpoint, creating a dataset for a real environment. Each environment was associated with datasets of varying sizes. The first and third rows of Figure 4a display observation sequences for the two datasets, while the corresponding action sequences are illustrated in the two rows of labels. During the training phase, each environment underwent continuous training using models with distinct structures. After a maximum of 100 epochs, all models achieved convergence. In the testing phase, for each model, we used a random sequences from the training phase to initialize memory through continuous prediction inference. Subsequently, we executed imagination to predict observations for the entire environment. The second and fourth rows of Figure 4a showcase the results of predictions. Figures 4b and 4e respectively demonstrate the predictive capabilities of the models for the environments post-training. As evident, an increase in the number of neurons correlates with enhanced predictive capabilities. Figures 4c and 4d illustrate that, with the same network size, larger datasets result in diminished model performance.

**Static observation while the environment changes.** When we take no action, remaining stationary, the external world can change continuously. For example, when watching a video, our observations constantly evolve. In such cases, we also need to predict future observations. We conducted evaluations using the MNIST-rot dataset and TaxiBJ dataset. The MNIST-rot dataset consists of

sequences with 20 frames each, while the TaxiBJ dataset sequences contain 8 frames. During training, we utilized $N = 128$ sequences. In testing, we use sequences that were not seen during training. For the MNIST-rot dataset, we initialized memory with the first 10 frames and then reproduced the entire sequence. For the TaxiBJ dataset, memory was initialized with the first 4 frames, and the entire sequence was reproduced from the beginning (Figure 6 in appendix). We achieved better results than tPCN on the MNIST-rot dataset (Table 2). And we compared our performance on the TaxiBJ dataset with BP-based machine learning models, achieving comparable results (Table 3).

|  | Seen | | Unseen | |
| --- | --- | --- | --- | --- |
| SeqLen | Ours | tPCN | Ours | tPCN |
| 16 | 3.4e-8 | 0.012 | 0.049 | 0.055 |
| 32 | 3.5e-7 | 0.018 | 0.041 | 0.045 |
| 64 | 2.7e-5 | 0.012 | 0.034 | 0.035 |
| 128 | 5.1e-4 | 0.011 | 0.022 | 0.023 |
| 256 | 0.002 | 0.011 | 0.017 | 0.018 |
| 512 | 0.003 | 0.009 | 0.011 | 0.017 |
| 1024 | 0.004 | 0.010 | 0.009 | 0.015 |

Table 2: MSE of predictions calculated on MNIST-rot dataset. We compare our model with tPCN Tang et al. (2023) for different sequence lengths.

| Model | Frame 1 | F2 | F3 | F4 |
| --- | --- | --- | --- | --- |
| ST-ResNet | 0.460 | 0.571 | 0.670 | 0.762 |
| VPN | 0.427 | 0.548 | 0.645 | 0.721 |
| FRNN | 0.331 | 0.416 | 0.518 | 0.619 |
| Ours | 0.458 | 0.514 | 0.567 | 0.633 |

Table 3: MSE of predictions calculated on TaxiBJ dataset. We compare our model with several popular methods in machine learning, including ST-ResNet Zhang et al. (2017), VPN Kalchbrenner et al. (2017), and FRNN Oliu et al. (2018). All compared models take 4 historical traffic flow images as inputs, and predict the next 4 images.

## 6 DISCUSSION

We consider that the brain employs an EBM as an intrinsic generative model to predict the next observation after action. We utilize a hierarchical neural network to implement this process and incorporate a CANN as memory to compress past experiences. As a biologically plausible neural network, our model succeeds in various environments with different actions. This provides insight into how the brain builds an internal model capturing the dynamics of the environment. Moreover, our model achieves performances on par with machine learning methods, indicating that our framework has the potential for further development.

Unlike previous machine learning works Hafner et al. (2020) or predictive coding network approaches Tang et al. (2023), our framework differs in that we first perform learning and then inference (see the order in Algorithm 1). In contrast to previous approaches that conduct learning after inference, this order in our framework leads to a distinct objective function. Our objective is expressed as $\mathbb{E}_{p(s|m)} \log p(o|s)$ (see Eq.(3)), whereas previous works often use $\mathbb{E}_{p_{\text{post}}} \log p(o|s)$. We experimented with the latter objective as well, but it resulted in poorer and less robust performance in our model. In fact, our approach of learning before inference is closer to the autoregressive models used in LLMs.

In the current model, when computing the gradient of the objective function with respect to the model parameters, we ignore the influence of parameters on memory (see Eq.(5)). This can be understood as a form of Truncated BPTT. Nevertheless, in experiments related to movement, we found that this does not affect our long-distance predictions.

**Future work.** Due to the Markovian nature of our generative model, our framework can seamlessly integrate with reinforcement learning (RL). By simply defining additional rewards for specific tasks, we can achieve model-based RL, creating a biologically plausible world model. In our current model, though direct access to the underlying state is not available, the utilization of CANNs effectively establishes a prior structure for the underlying state. We have not yet thoroughly investigated the impact of the environment dynamics on the CANN structure. The two for-loops in Algorithm 2 are executed sequentially. In neural systems, however, all neurons compute simultaneously. It has been demonstrated that by introducing a modulation function for error neurons, both for-loops can be unrolled to achieve parallel computation Song et al. (2020). Nevertheless, when using a real continuously changing environment, we still need to carefully adjust the model's time constants to ensure it can maintain synchronized interaction with the real environment. We will explore these aspects in our future work.

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

## A  LOWER BOUND DEVIATION 1

Firstly, we can prove that,

$$p(o|s)\log p(o|s) - \log p(o|s), \tag{19}$$
$$= [p(o|s) - 1]\log p(o|s), \tag{20}$$
$$\geq 0 \tag{21}$$

Then, the mutual information between sample $o \sim p_{\text{true}}(o)$ and $s$ under $p$ distribution is calculated as,

$$\mathbb{E}_{o \sim p_{\text{true}}(o)} I(o, s|m) = \mathbb{E}_{o \sim p_{\text{true}}(o)} \mathbb{E}_{s,m \sim p(o,s,m)} \log \frac{p(o, s, m)p(m)}{p(o, m)p(s, m)}, \tag{22}$$

$$= \mathbb{E}_{o \sim p_{\text{true}}(o)} \mathbb{E}_{s,m \sim p(o,s,m)} \log \frac{p(o, s, m)}{p(s, m)} + \mathbb{E}_{o \sim p_{\text{true}}(o)} \mathbb{E}_{m \sim p(m)} \log \frac{p(m)}{p(o, m)} \tag{23}$$

$$= \mathbb{E}_{o \sim p_{\text{true}}(o)} \mathbb{E}_{s,m \sim p(o,s,m)} \log p(o|s) + \text{Const}, \tag{24}$$

$$\overset{+}{=} \mathbb{E}_{o \sim p_{\text{true}}(o)} \mathbb{E}_{m \sim p(m)} \mathbb{E}_{p(s|m)} p(o|s) \log p(o|s), \tag{25}$$

$$\geq \mathbb{E}_{m \sim p(m)} \mathbb{E}_{o \sim p_{\text{true}}(o)} \mathbb{E}_{p(s|m)} \log p(o|s). \tag{26}$$

The relation between $\mathcal{H}$ and $\mathcal{L}$ is written as,

$$\mathcal{L} = \mathcal{H} + \mathbb{E}_{o \sim p_{\text{true}}(o)} D_{\text{KL}}\left[p(s|m)||p_{\text{post}}\right]. \tag{27}$$

## B  LOWER BOUND DEVIATION 2

The mutual information between $o$ and $s$ under $q$ distribution is calculated as,

$$I(o, s|m) = \mathbb{E}_{o,s,m \sim q(o,s,m)} \log \frac{q(o, s, m)q(m)}{q(o, m)q(s, m)}, \tag{28}$$

$$= \mathbb{E}_{o,s,m \sim q(o,s,m)} \log \frac{q(o, s, m)}{q(s, m)} + \mathbb{E}_{o,m \sim q(o,m)} \log \frac{q(m)}{q(o, m)}, \tag{29}$$

$$= \mathbb{E}_{o,s,m \sim q(o,s,m)} \log q(o|s) + \text{Const}, \tag{30}$$

$$\overset{+}{=} \mathbb{E}_{o,s,m \sim q(o,s,m)} \log q(o|s), \tag{31}$$

$$\geq \mathbb{E}_{o,s,m \sim q(o,s,m)} \log q(o|s) - \mathbb{E}_{s \sim q(s,m)} D_{\text{KL}}\left[q(o|s)||p(o|s)\right], \tag{32}$$

$$= \mathbb{E}_{o,m \sim q(o,m)} \mathbb{E}_{s \sim q(s|o,m)} \log p(o|s). \tag{33}$$

$$\tag{34}$$

## C  THE CANN DYNAMICS

We use $m_t \in \mathbb{R}^{n_L}$ to represent the firing rate of the CANN at time $t$, and $I_t \in \mathbb{R}^{n_L}$ to represent the total synaptic input to the neurons in CANN. According to the integral firing model, the firing rate can be approximated as,

$$m_t = H(I_t), \tag{35}$$

where $H(\cdot)$ is a nonlinear function. The dynamics of $I_t$ are determined by its own relaxation, recurrent inputs from other neurons, neural adaptation $V_t$, and external inputs from $s_t^L$ and action neurons, as expressed by the following equation:

$$\tau_I \frac{dI_t}{dt} = -I_t + Wm_t - V_t + s_t^L + a_t \tag{36}$$

Here, $\tau_I$ represents the synaptic time constant, and $W$ denotes the recurrent neuronal connections. $W$ is a randomly generated low-rank matrix, where its norm is controlled by the hyperparameter $\alpha = \|W\|$. The dynamic of $V_t$ is written as,

$$\tau_V \frac{dV_t}{dt} = -V_t + \beta m_t \tag{37}$$

where $\tau_V$ is the adaptation time constant and $\beta$ is a scalar controlling the adaptation strength.

## D    SUPPLEMENTARY FIGURES

Figure 5a illustrates the learning process of the eye movement experiment. Figure 5b presents clearer generated images, while Figures 5c and 5d depict the changes in MSE and layer losses during the model training process in the eye movement experiment. Figure 6 showcases the experimental results from static observations as the environment changes.

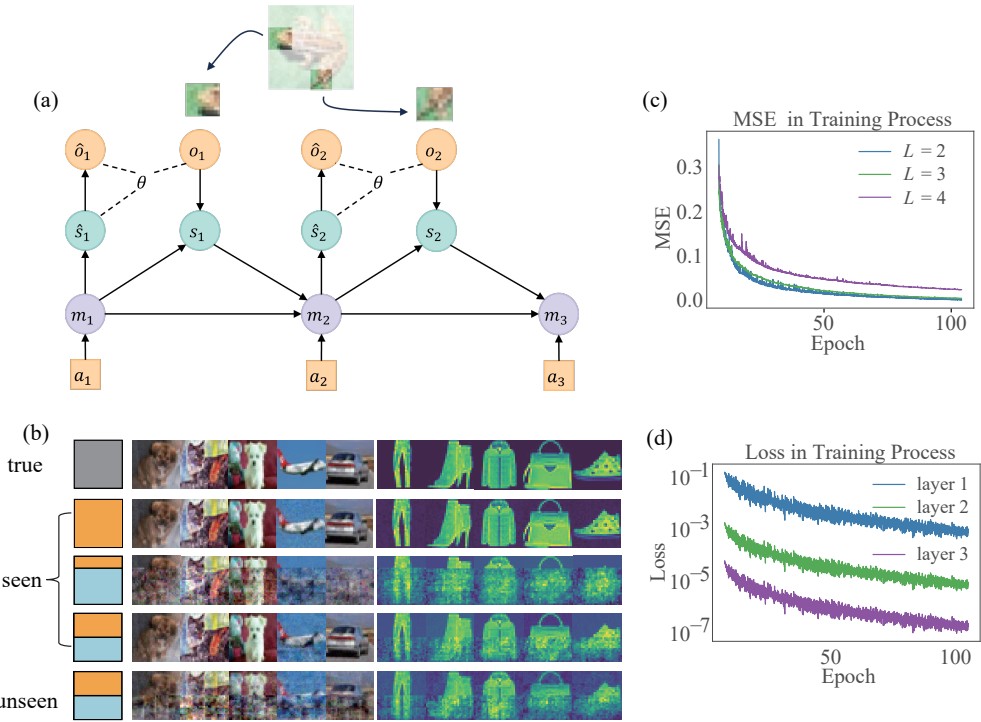

Figure 5: (a) The learning process of the eye movement experiment. (b) Supplement to Figure 3a in the main text. (c) The dataset used here is CIFAR-10, with a total neuron count of $\sum_l n_l = 3\text{k}$ being consistent across models with different layers $L$. (d) The loss decreases exponentially as the number of layers increases, with the model trained on $N = 64$ images.

## E    HYPER PARAMETERS

Here are the hyperparameters used in the experiment. All programs are run on one NVIDIA RTX A6000, and we use JAX (CUDA 11) to accelerate the programs. For the experiments depicted in our figures, each one takes 5-20 minutes, with the best performance on the DeepLab and Google Street datasets requiring about 10 hours. The code will be open-sourced after publication.

For all stochastic differential equations, we employ the Euler method for simulation with a step size of $dt$. Both inference and learning after a single observation were conducted over a total simulation time of $T$. In other words, for a time step from $t$ to $t + 1$, the simulation occurs $T/dt$ times. The duration of operation for each layer is uniform. All models use the leaky_relu activation function denoted as $f(\cdot)$. The parameter $\tau_s, \tau_I$ and $\tau_V$ are set to 1 in all models.

In Tables 2 and 3, we used the corresponding parameter settings from the original texts. For the TransDreamer in Table 1, the parameter settings are as follows:

- Attention head = 8, Dropout = 0.2, Hidden size = 128, Model size = 64, Layers = 3

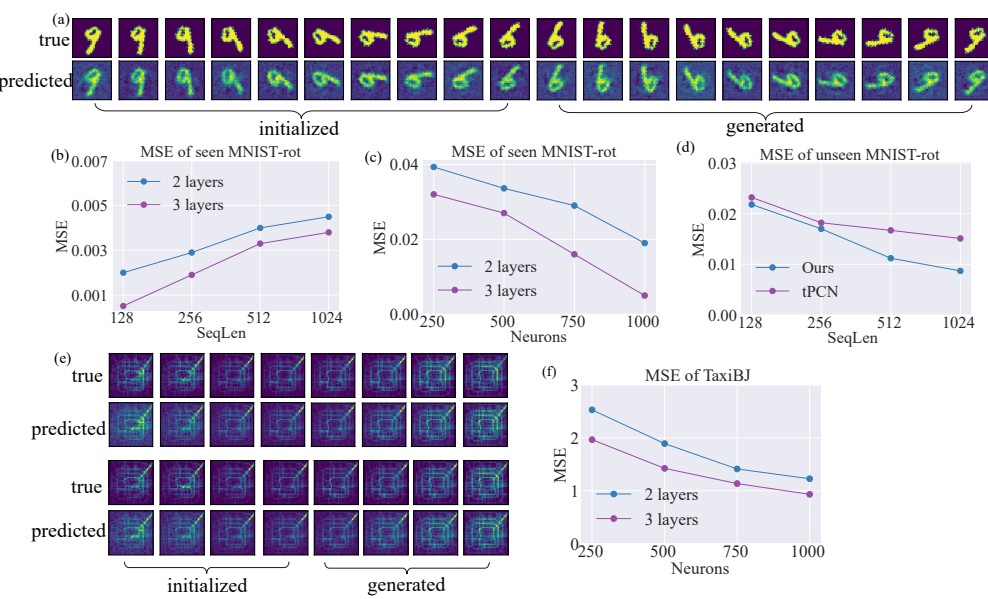

Figure 6: Experiments on sequential data. (a) The 1st row shows the ground truth, and the 2nd row presents the predicted results. We use $K = 10$ frames to initialize memory in the MNIST-rot dataset. (b)(c) Detailed experiments on the MNIST-rot dataset under various parameters. (d) Comparison results with tPCN. (e) Rows 1 and 3 correspond to the ground truth, while Rows 2 and 4 represent the predictions generated by the model. We use $K = 4$ frames for the TaxiBJ dataset. (f) Detailed experiments on the TaxiBJ dataset under different parameters.

| DATASET | $n_0$ | $L$ | $n_l \ (l > 0)$ | d$t$ | $T$ | $1/\tau_\theta$ | $\alpha$ | $\beta$ | EPOCHS |
|---------|-------|-----|-----------------|------|-----|-----------------|----------|---------|--------|
| FASHION-MNIST | $7 \times 7$ | 2 | 1500,1500 | 0.05 | 10 | 0.1 | 1 | 0 | 125 |
| | | 3 | 1000,1000,1000 | | | | 0.5 | | |
| | | 4 | 750,750,750,750 | | | | 0.1 | | |
| CIFAR-10 | $3 \times 8 \times 8$ | 2 | 1500,1500 | 0.05 | 10 | 0.1 | 1 | 0 | 125 |
| | | 3 | 1000,1000,1000 | | | | 0.5 | | |
| | | 3 | 512,256,128 | | | | 0.5 | | |
| | | 4 | 750,750,750,750 | | | | 0.1 | | |
| DEEPMIND LAB | $3 \times 80 \times 60$ | 3 | 1000,1000,1000 | 0.05 | 10 | 0.01 | 0.5 | 0 | 40 |
| | | 4 | 750,750,750,750 | 0.05 | | | 0.1 | | 40 |
| | | 4 | 4000,2000,2000,2500 | 0.02 | | | 0.1 | | 100 |
| GOOGLE STREET | $3 \times 100 \times 50$ | 3 | 1000,1000,1000 | 0.05 | 5 | 0.1 | 0.5 | 0 | 40 |
| | | 4 | 750,750,750,750 | 0.05 | 7 | 0.1 | 0.1 | | 40 |
| | | 4 | 4000,2000,2000,2000 | 0.1 | 5 | 0.05 | 0.1 | | 140 |
| MNIST-ROT | $28 \times 28$ | 3 | 2000,1000,1000 | 0.05 | 10 | 0.1 | 0.5 | 1 | 100 |
| TAXIBJ | $32 \times 32$ | 3 | 2000,1000,1000 | 0.05 | 10 | 0.1 | 0.5 | 1 | 200 |

Table 4: Parameters setting for different models

