# OpenReview forum: "Predicting Observation after Action in a Hierarchical Energy-based Model with Memory"
_ICLR.cc/2025/Conference — Submitted to ICLR 2025_

### Official Review · Reviewer_DMCv · 2024-10-22

**Soundness:** 2
**Presentation:** 3
**Contribution:** 2
**Rating:** 3
**Confidence:** 4

**Summary:**

This paper proposes an energy-based neural recurrent state-space model designed to predict world perception following an action. While prior works have introduced world models for predicting observations, they often rely on techniques like Backpropagation Through Time (BPTT), which lack biological plausibility. In contrast, this paper employs an energy-based state-space model, which is known to resemble biological mechanisms such as Hebbian learning and sampling-based inference, to address the sequential prediction task. The model is implemented using a hierarchical neural network for its expressiveness and biological validity, and a continuous attractor network is utilized to model temporal memory, similar to biological neural systems. Evaluation on a few vision-based tasks demonstrates that the model outperforms existing techniques in predicting observations.

**Strengths:**

1. The paper provides a detailed formulation of each component and effectively connects them to biological concepts, supporting its claim of biological plausibility.
2. The authors successfully integrate several neural network-based components to simulate a biological system plausibly.
3. The paper is well-written overall, with clear visualizations that enhance understanding.

**Weaknesses:**

1. The main technique does not appear to offer significant practical advantages beyond its biological plausibility.
2. While the paper argues that the learning rule is biologically plausible due to its resemblance to the Hebbian rule (as opposed to BP), the formulation and its properties remain quite similar to BP, which diminishes the claimed advantage of biological plausibility.
3. The evaluation is limited; the setups simulate abstract and simplistic biological behaviors, without testing the model on more realistic and practical tasks.

**Questions:**

I am open to adjusting my scores if the following comments are addressed:
- Please add parentheses around the citations, as the current formatting makes them difficult to read.
- While the world/observation prediction task is prevalent in biological systems, what practical applications does this have? Does it improve the performance of sequential decision-making algorithms? A more application-driven empirical evaluation would be necessary.
- It would be beneficial to include evaluations on more practical decision-making tasks that could benefit from this model, particularly by solving sub-tasks related to world prediction. Although this is mentioned in the discussion, the practical significance of the technique is unclear without actual end-to-end demonstration.
- The paper's practical motivation is not entirely convincing. Using local learning rules or neural dynamics instead of BP is argued to be advantageous, but what practical benefits does it offer? BP is known for its high memory usage and additional latency, but how does the local learning rule address these issues?
- The claim that the learning rule differs significantly from BP, or that it fully matches the Hebbian rule, is unclear. In line 281, the paper states, "Each layer will repeat this process and propagate information downward until the last layer." While it is true that gradients are not propagated backwards through the hierarchical network, latent information $s^l_t$​ is still propagated backwards. This necessitates storing all intermediate activations and backpropagating the latent information, much like conventional BP. The only aspect that aligns with the Hebbian rule is the first layer. Additionally, in Equation 16, the update rule for the sample $s^l_t$ involves the derivative $f'$ and the derivative of the Gaussian reparametrization, and this updated sample is used to update the subsequent layer. This resembles backpropagating gradient information through the sample updates. A dedicated subsection verifying that the suggested formulations are indeed biologically plausible would strengthen the paper. For example, it would be useful to mathematically verify that the learning rules adhere to the Hebbian rule and differ substantially from backpropagation. Since biological plausibility is a key advantage of the proposed technique, a thorough theoretical analysis would be necessary.
- In the imagination phase, to predict the outcome of a specific action within the observed world, must all previous sequences of actions and observations be sequentially inputted to form the memory network? Is there a component where one can input the initial observation directly?
- Why is image quality compared using MSE when there are several established image quality metrics, such as PSNR or SSIM, that could be used? MSE seems ambiguous in this context, and it would be preferable to employ commonly accepted metrics.
- The evaluation setups may need further validation to confirm that they correctly simulate biological behaviors and perceptions. For example, is it valid to simulate eye movement with datasets like CIFAR-10 or FMNIST? I would expect a more complex real-world task beyond 2D image scanning. Is there biological literature that supports the use of such datasets to simulate eye movements?
- What are the practical advantages of this energy-based model, aside from its biological plausibility? The performance does not seem to scale with the number of trained images, which raises concerns about its practicality.
- Several minor corrections:
  - Line 154: "decent" should be "descent"
  - Line 186: "deviation" should be "derivation"
  - Line 209: "employs" should be "employ"
  - Line 334: "first is step is" should be "first step is"

---

> ### Author Response · Authors · 2024-11-21
>
> We sincerely thank you for your thorough review and constructive suggestions. Your insights have helped us better articulate our contributions and refine our work. Below, we address your concerns in detail.
>
> **Weaknesses:**
> 1. We would like to emphasize that the biological plausibility of our work is a key focus, aiming to explain how the brain performs representation learning. Utilizing biologically plausible networks to interpret brain mechanisms contributes significantly to the community, as demonstrated by works such as Tommaso Salvatori (2021) and Yuhang Song (2023). Moreover, our brain-inspired model achieves performance comparable to BP-based models on several benchmarks used in our experiments. Among non-BP methods, our approach represents one of the best-performing algorithms on similar benchmarks.
> 2. There are differences between our method and BP in terms of implementation. For detailed explanations, please refer to Question 5.
> 3. Predictive learning, as a form of representation learning, is already a mature upstream task. Our focus lies on predictive learning itself, and downstream tasks are beyond the scope of this paper.
>
> **Questions:**
> 1. In the revised version, we will use a clearer citation format.
>
> 2-3. Predictive learning is inherently a type of representation learning, aimed at learning representations that facilitate prediction. This is already a critical area of interest, with many works focusing on predictive learning itself, such as Mufeng Tang (2023) (biologically plausible models), Aaron van den Oord (2019), Tung Nguyen (2021), and Mufeng Tang (2023) (BP-based models). As a general representation learning task, predictive learning supports a variety of downstream applications, including model-based RL and video generation. Our focus is on predictive learning as a representation learning task, and downstream tasks are beyond the scope of this paper.
>
> 4. One of the focal points of this work is understanding how the brain performs representation learning. By employing a biologically plausible model based on neural dynamics and local learning, we provide a framework to explain this mechanism, which constitutes a major contribution of this study.  Additionally, our brain-inspired model achieves performance comparable to BP-based models on several benchmarks. While our algorithm does not yet show significant advantages in memory usage or latency over BP when implemented on von Neumann computers, it is well-suited for in-memory computing chips. These chips significantly reduce data transmission latency and bandwidth issues, offering a promising potential advantage for our algorithm over BP.
>
> 5. Thank you for thoroughly reviewing the details of our algorithm. While our method shares similarities with BP in terms of hierarchical information transfer, it differs substantially in its derivation and implementation. In our model, each layer contains not only the state neurons ($s$) but also additional error neurons ($e$). The joint operation of $e$and $s$ in each layer facilitates posterior sampling and gradient computation. BP, by contrast, relies solely on \(s\) without involving error neurons. Furthermore, as each layer has error neurons, the synaptic updates in our model adhere to the Hebbian rule. The update rule for the first layer is presented in Eq. 15, while the subsequent layers follow the formula:
> $\tau_{\theta}\frac{\mathrm{d}\theta^l}{\mathrm{d}t} = -\nabla_{\theta^l}\mathcal{L}^{l}_t = \Lambda^l\hat{e}^l_t f(\hat{s}^{l+1}_t)^T.$
> We will include this equation in the revised version.
> 6. Since our algorithm operates online, it cannot initialize the model with all past sequences at once; instead, it processes data sequentially.
> 7. We used MSE as the evaluation metric to facilitate comparison with prior work in this domain, such as tPCN, VPN, and ST-ResNet, all of which also adopted MSE.
> 8. For the CIFAR-10 dataset, we split each image into 16 patches to simulate the receptive field of real eye movements. In future work, we plan to use more realistic images to mimic eye-tracking experiments.
> 9. Thank you for the question. The performance of our model on the test dataset scales with the number of neurons, as shown in Table 1 and Figure 3.e. The observed degradation on seen images (training set) with increased neuron numbers is due to the increased difficulty of initialization. Energy-based models (EBMs) exhibit strong learning capabilities for both continuous and discontinuous functions in representation learning, whereas traditional forward models such as VAE face challenges with discontinuous functions. Please refer to *Implicit Behavioral Cloning* for more details.
> 10. Thank you for the suggestion.
>
> We hope this explanation addresses your concerns and provides clarity regarding our work.

---

### Official Review · Reviewer_9CDg · 2024-10-26

**Soundness:** 2
**Presentation:** 1
**Contribution:** 2
**Rating:** 3
**Confidence:** 4

**Summary:**

The authors build a series of predictive neural networks following the energy-based model. What is particularly interesting about the model is the incorporation of several key ingredients: attractor network / memory, predictive coding, and a hierarchical structure. It is difficult to assess what question(s) the authors are trying to address. The abstract and introduction are a little bit all over the map, repeatedly referring to “biological plausibility”, a fashionable term that ends up being rather empty without any anchoring on actual biological data. The work also refers to eye movements, and motion in environments, both of which are only superficially discussed at a very abstract level.

**Strengths:**

There are many interesting ideas in here, including: (1) the energy-based models and their relationship to predictive coding, (2) the emphasis on predicting actvitiy after an action is taken, (3) neural network architectures that have a memory component, recurrent dynamics, hierarchies, and prediction.

**Weaknesses:**

The work would benefit a lot from being more specific throughout. Take the abstract, which continuously refer to “the neural system”. What does this mean? Is this referring to any arbitrary neural network? A fly? A human? Theories of everything often end up being theories of nothing. What are the authors trying to model here?

It would be very useful to articulate a key question or set of questions or hypotheses. Are the authors trying to build a network that can solve a specific task?
If so, which task, what are the benchmarks, what are other comparison models?
If they are not trying to solve a specific task, is there a concrete hypothesis that the authors are testing?

To the extent that this work attempts to connect with biological brains (it is not clear whether this is the case), a lot of work would be needed to establish those connections. Take the first sentence of the discussion: “We consider that the brain employs an EBM as an intrinsic generative model to predict the next observation after action.”. Is this a hypothesis, a conjecture, a conclusion, or an axiom? If it is a hypothesis, then what kind of *empirical* data would be needed to test the hypothesis? Are there any neuroscientific data that provide support for this hypothesis or evidence that contradicts the hypothesis?

**Questions:**

It seems that one of the interests is to model eye movements (curiously referred in the singular as eye movement throughout). There is a whole industry or real neural network models that aim to predict eye movements in images under different circumstances. Here are a few: Zhang et al Nature Communications 2018, Yang et al CVPR 2020 (these are studies during specific actions for visual search), Kummerer et al J Vision 2022 (this is without any task). In the absence of comparisons with actual eye movements, it is not clear how to evaluate the work here. But perhaps the intent is not truly to study eye movements, the actual question or goal was not clear.

The notion of “imagination” and being able to predict observations for actions without performing the actions seems quite interesting. It would be quite interesting to actually show experimental data on this. It seems that the experiments in Fig.3 (“eye movement”) and Fig. 4 (“head turning”) are in this direction. What were the models trained on and how was cross-validation performed? Can the model take a novel image and make inferences about observations for a novel viewpoint never seen before? What is “ture” in Fig. 4?

As a general note, to interpret results, I always find it very useful to include:
-	Error bars.
-	Chance levels.
-	Upper bounds (not entirely sure how to define it here, but perhaps an oracle version)
-	Rigorous statistics.
This comment applies to all tables and Figs. 3. and 4.

---

> ### Author Response · Authors · 2024-11-21
>
> Thank you for taking the time to review our manuscript and for providing thoughtful feedback. Below, we address each of the concerns and questions you raised.
>
> **Weakness**:
>
> 1. To clarify, our work assumes that the brain uses a predictive learning paradigm to perceive and interact with the world. Based on this assumption, we propose a biologically plausible network for predictive learning-based representation learning. The brain processes information through different pathways depending on the sensory-motor system involved. Our model aims to simulate these neural pathways. For example, in our saccadic eye movement experiments, hierarchical EBMs model the visual cortical pathway (V1-V4-IT neocortex), while CANNs simulate the collicular pathway for motor control. Similarly, in our movement experiments, EBMs model the visual cortical pathway, and CANNs represent grid cells in the entorhinal cortex. Since our work primarily focuses on the modeling aspects, we did not elaborate on specific cortical correspondences in the manuscript.
>
> 2. The specific task we address is *predictive learning*. Predictive learning is a form of representation learning aimed at acquiring representations optimized for prediction, making it an inherently valuable and widely researched topic. Many works have focused on predictive learning, such as Mufeng Tang (2023, biologically plausible model algorithms) and Aaron van den Oord (2019, BP-based algorithms). Predictive learning is a general and versatile task for representation learning. In our study, we compared our approach with five other models, presenting results across three benchmark datasets, as shown in Tables 1, 2, and 3.
>
> 3. Our work is based on the widely adopted Bayesian brain hypothesis, as discussed by Knill & Pouget (2004) and Friston & Price (2001), which posits that the brain understands the world through predictive learning. This hypothesis has inspired a significant body of work, including Gergo Orbán (2016) and Guillaume Hennequin (2014). Furthermore, it is supported by experimental evidence from studies such as Ulrik R. Beierholm (2009), Edward H. Nieh (2021), and Ralf M. Haefner (2016).
>
> ---
>
> **Questions**:
>
> 1. Thank you for highlighting this point. To clarify, our task is not to predict the saccadic eye movement itself but rather to predict the post-saccadic image based on the known eye movement position.
>
> 2. In Figure 3a, "unseen" images refer to novel images that were not encountered during training. The models used in this row were trained on datasets that excluded the specific images shown. In Figure 4, "true" refers to the ground truth, i.e., the actual images.
>
> 3. Thank you for your suggestion. The shaded regions in Figure 3 represent the error bars.
>
> We hope this explanation addresses your concerns and provides clarity regarding our work.

---

> > ### Comment · Reviewer_9CDg · 2024-11-22
> > **Vague**
> >
> > The manuscript and responses continue on the same tone of vagueness.
> > Starting at the end, "The shaded regions in Figure 3 represent the error bars." What kind of error bars? Standard deviations? 95% Confidence intervals? Other? This is but one simple example of the requirement for specificity throughout.
> >
> > If the goal is not to predict eye movements but rather to "predict the post-saccadic image", what does this mean? Is the idea here that the authors have a model of eccentricity-dependent sampling with receptive field sizes that are progressively larger from one area to another and progressively larger from the fovea to the periphery? Where and how is this implemented?
> >
> > "... the brain uses a predictive learning paradigm ..."
> > Which brain? C.elegans? Humans? Ants? All?
> > Assuming that the authors are interested in mammalian brains and perhaps even primate brains, which brain areas? The cerebellum? The retina? Prefrontal cortex? All of them?
> > Assuming that the authors are interested in cortex, which cortical layer? All of them?
> > Which neuronal types? All of them?
> > Such generic statements as the ones used throughout the paper and in this response are very hard to connect with actual data.
> >
> > I suggest that the authors think about a major rewrite being extremely specific about what the hypothesis are, what hte questions are, which brains and which particular cognitive functions they are interested in modeling.

---

> > > ### Author Response · Authors · 2024-11-24
> > >
> > > Thank you for your review. The "shaded regions" here, as in the conventional usage, refer to standard deviations, similar to the error bars in Fig. 2 of Ref. [1], which are not explicitly labeled. We believe it is generally understood that these represent standard deviations. In any case, we will make sure to clarify this point in the revised version of the paper. Thank you for your suggestion.
> > >
> > > The eye movement experiment is designed to demonstrate how predictive learning can be implemented using an eye movement paradigm, where the target of the predictive learning is to "predict the post-saccadic image."
> > >
> > > Our research focuses on the nervous systems of vertebrates, including common model organisms such as rats, macaques, mice, and humans. The brain structures in these vertebrates are relatively similar, which is why we did not specify them. For details on the different sensory-motor pathways and their associated brain regions, please refer to the response above. In fact, current studies on brain-inspired networks do not typically specify the model organisms or the corresponding brain regions, as seen in Refs. [1-4].
> > >
> > > We hope our responses can address your concerns.
> > >
> > > [1]  Dong, Xingsi, et al. "Adaptation accelerating sampling-based bayesian inference in attractor neural networks." *Advances in Neural Information Processing Systems* 35 (2022): 21534-21547.
> > >
> > > [2] Tang, Mufeng, Helen Barron, and Rafal Bogacz. "Sequential memory with temporal predictive coding." *Advances in neural information processing systems* 36 (2024).
> > >
> > > [3] Salvatori, Tommaso, et al. "Associative memories via predictive coding." *Advances in Neural Information Processing Systems* 34 (2021): 3874-3886.
> > >
> > > [4] Zhang, Wen-Hao, et al. "Sampling-based Bayesian inference in recurrent circuits of stochastic spiking neurons." *Nature communications* 14.1 (2023): 7074.

---

### Official Review · Reviewer_rQhX · 2024-11-03

**Soundness:** 3
**Presentation:** 3
**Contribution:** 2
**Rating:** 5
**Confidence:** 2

**Summary:**

This paper proposes a hierarchical recurrent state-space model based on an energy-based model (EBM) to simulate the brain's process of predicting observations after actions. By introducing a continuous attractor neural network (CANN) for memory, the paper constructs a biologically plausible neural network model. The model is validated across various tasks, including vision, motion, and head rotation, demonstrating its predictive performance in both trained and novel environments. Based on a bio-inspired framework, this model shows comparability to existing machine learning methods.

**Strengths:**

1.Bio-inspired Modeling: The model leverages the EBM framework and CANN to simulate prediction, learning, and inference in the neural system, achieving biological plausibility, which is a unique attempt in machine learning.

2.Innovative Prediction and Learning Sequence: Unlike most machine learning models, this model performs learning before inference, resulting in a unique objective function. This sequence is shown to provide greater stability and robustness over traditional methods.

3.Memory Mechanism with CANN: The integration of CANN as a memory network allows the model to efficiently compress past experiences, supporting real-time prediction.

**Weaknesses:**

1.Biological Interpretability of Experimental Results: Although the model shows good predictive performance, its interpretability in terms of biological neural systems could be strengthened. For instance, explaining how error neurons represent biological neuron behavior or how model layers correspond to cortical structures would enhance understanding.

2.Comparison of Baselines: While the paper includes comparisons with models like TransDreamer, it would be beneficial to compare with more bio-inspired models (e.g., more complex PCN frameworks) to further highlight the model's advantages in biological plausibility and performance.

3.In-depth Analysis of Model Parameters: Model performance is likely influenced by parameters like network depth and neuron count. However, the paper lacks a detailed discussion of how these parameters impact biological plausibility. Future work could examine the model’s adherence to biological realism under various parameter settings.

4.Impact of Dynamic Environment Changes on CANN: The framework does not fully analyze how dynamic environmental changes affect the CANN structure. Although the paper mentions this as future work, adding preliminary exploration in current experiments could help validate CANN’s stability in complex environments.

**Questions:**

1.Future versions could provide more detail on how the model’s biological mechanisms correspond to actual neural processes, such as simulating firing patterns and memory representations of biological neurons.

2.Consider additional experiments with various parameter settings to analyze the model’s adaptability and robustness in more complex environments, especially in comparison to traditional bio-inspired models.

3.Further discussions could be added regarding the representational role of CANN in brain structures and the correspondence between the model’s hierarchical layers and cortical structure.

---

> ### Author Response · Authors · 2024-11-20
>
> Thank you for acknowledging our work on achieving biological plausibility，Innovative Prediction and Learning Sequence that bring greater stability and robustness over traditional methods, and validating the effectiveness of our memory mechanism. We hope the following responses can address your concerns:
>
> **W1, Q1, Q3:** We appreciate your recognition of our model's good predictive performance. The error neurons employed in our work are supported by extensive experimental evidence. For example:
> - [1] successfully explained end-stopping and other extra-classical receptive-field effects in the primary visual cortex by incorporating error neurons.
> - [2–3] provided evidence for functionally distinct neuronal subpopulations, potentially corresponding to prediction and error neurons.
> - [4] discussed multi-compartment neurons with a distinct apical dendrite compartment for storing prediction errors separately from the value encoded by the soma.
> - [5] introduced a central microcircuit model of predictive processing using error neurons.
> - [6] described predictive processing as a computational framework for cortical function, particularly in sensory processing.
> - [7] extended formulations of the brain as an inference organ, using error neurons to orchestrate these processes.
> - [8–12] explained a range of perceptual and neural phenomena, including repetition suppression, error responses, attentional modulations, bistable perception, and motion illusions.
>
> We aim for our model to represent a general mechanism for the brain's predictive functions. For instance, in the context of vision, the hEBM framework in our model aligns with the hierarchical structure of the visual cortex. Similarly, CANN, as the memory module, has robust biological support, particularly in the hippocampus. Examples include place cells [13–15], saccadic movement position cells [16], head direction cells [17], movement direction cells [18], and orientation cells [19].
>
> **W2, Q2:** In Table 2, we compare our model with the recent tPCN framework. To the best of our knowledge, our network is the first biologically plausible model capable of predicting different observations based on different actions. For instance, general PCN frameworks can only handle static tasks, and tPCN extends this to dynamic tasks but only with fixed input sequences. Our network can produce different sequences based on distinct actions.
>
> **W3** Figures 3c, 4b–e, and 6b–d in the paper demonstrate how model performance is influenced by network parameters and neuron counts. The question of how these parameters impact biological plausibility is intriguing. In future work, when applying the network to specific cortical regions, we plan to set these parameters based on real biological data.
>
>  **W4** As mentioned in the discussion section, this is an interesting question. We intend to explore it in future research, although it is beyond the scope of the current paper.
>
> ref:
> [1] Rao, R. and Ballard, D. "Predictive coding in the visual cortex." Nat. Neurosci., 1999.
>
> [2] Bell, A. H., et al. "Encoding of stimulus probability in macaque inferior temporal cortex." Curr. Biol., 2016.
>
> [3] Fiser, A., et al. "Experience-dependent spatial expectations in mouse visual cortex." Nat. Neurosci., 2016.
>
> [4] Takahashi, N., et al. "Active dendritic currents gate descending cortical outputs in perception." Nat. Neurosci., 2020.
>
> [5] Bastos, A. M., et al. "Canonical microcircuits for predictive coding." Neuron, 2012.
>
> [6] Keller, G. B., and Mrsic-Flogel, T. D. "Predictive processing: A canonical cortical computation." Neuron, 2018.
>
> [7] Kanai, R., et al. "Cerebral hierarchies: Predictive processing, precision, and the pulvinar." Phil. Trans. R. Soc. B, 2015.
>
> [8] Auksztulewicz, R., and Friston, K. "Repetition suppression and its contextual determinants in predictive coding." Cortex, 2016.
>
> [9] Feldman, H., and Friston, K. J. "Attention, uncertainty, and free-energy." Front. Hum. Neurosci., 2010.

---

> ### Author Response · Authors · 2024-11-20
> **Part2 ref**
>
> [10] Weilnhammer, V., et al. "A predictive coding account of bistable perception." PLoS Comput. Biol., 2017.
>
> [11] Lotter, W., et al. "A neural network trained to predict future video frames mimics biological neuronal responses." arXiv, 2018.
>
> [12] Watanabe, E., et al. "Illusory motion reproduced by deep neural networks." Front. Psychol., 2018.
>
> [13] Samsonovich, A., and McNaughton, B. L. "Path integration and cognitive mapping in a continuous attractor neural network model." J. Neurosci., 1997.
>
> [14] McNaughton, B. L., et al. "Path integration and the neural basis of the cognitive map." Nat. Rev. Neurosci., 2006.
>
> [15] Burak, Y., and Fiete, I. R. "Accurate path integration in continuous attractor network models of grid cells." PLoS Comput. Biol., 2009.
>
> [16] Seung, H. S. "How the brain keeps the eyes still." Proc. Natl. Acad. Sci. U.S.A., 1996.
>
> [17] Zhang, K. "Representation of spatial orientation by the intrinsic dynamics of the head direction cell ensemble."
>
> [18] Georgopoulos, A. P., et al. "Neuronal population coding of movement direction." Science, 1986.
>
> [19] Ben-Yishai, R., et al. "Theory of orientation tuning in visual cortex." Proc. Natl. Acad. Sci. U.S.A., 1995.

---

### Official Review · Reviewer_EQBX · 2024-11-06

**Soundness:** 3
**Presentation:** 3
**Contribution:** 3
**Rating:** 6
**Confidence:** 2

**Summary:**

The submission posted a solution an interesting topic on the neural mechanism to perform prediction of future observations given current state and memory. Authors proposed an energy based model to preform prediction, inference, and learning. A hierarchical and continuous attractor network is used to achieve these goals. Experimental results show the model can faithfully generate future observations for the environments the model is trained on and for unseen environments.

**Strengths:**

The introduced energy-based recurrent state space and a continuous attractor network to conceptually mimic the brain mechanism to predict next observations is interesting. The methodology introduction is clear and well organized.

**Weaknesses:**

The generalization ability of the proposed method is difficult to evaluate, as in Fig. 3a, the "unseen" images appear very similar to the previously "seen" images. It would be helpful to provide the mean squared error (MSE) between these unseen and seen images to clarify the degree of generalization.

What is the computation complexity for layered EBM structure and CANN memory integration. For a comprehensive comparison with baseline methods, it would be beneficial to discuss computational efficiency and scalability for larger datasets.

Additionally, the specific contribution of each component should be discussed with ablation studies. The proposed method involves several components—energy-based modeling, the continuous attractor network, and the hierarchical neural network—but it remains unclear which components contribute most significantly to prediction accuracy.

**Questions:**

1. The motivation for using an energy-based model is biological plausibility. However, it would be beneficial if the authors could discuss how the proposed model architecture improves prediction purely from a methodological/application perspective.

2. Table 3. how about comparison with more recent methods.

3. How the number of hierarchical layer affects the prediction accuracy.

---

> ### Author Response · Authors · 2024-11-20
>
> Thank you for recognizing our work as posting a solution an interesting topic on the neural mechanism to perform prediction of future observations, as well as for acknowledging the value of our experimental results. We hope the following responses can address your concerns:
>
> **W1.** The unseen images and seen images are chosen from the same dataset to specifically demonstrate the generalization ability of the model. For the experiments on unseen images, the network was trained on different images, not the ones shown in the demonstration. Below is the relationship between the MSE for unseen images and image numbers.
> Here’s how you can present the table and its description in English:
>
>
> | **Image Numbers** | 16     | 32     | 64     | 128    |
> |--------------------|--------|--------|--------|--------|
> | **MSE**           | 0.0682 | 0.0646 | 0.0613 | 0.0589 |
>
>
> **W2.** The computational complexity of our network at each iteration before convergence is $O(n^2 \times L)$, where $L$ represents the number of network layers, and $n$ represents the number of neurons per layer. As a result, the scalability of our network on larger datasets is consistent with that of a typical feedforward network.
>
> **W3.** The main contribution of our work lies in leveraging a fully biologically plausible network to achieve prediction tasks that the brain must perform. Specifically, hEBM is used for spatial compression, while CANN enables temporal compression. Removing certain components would render these processes biologically implausible within the neural system. For instance, removing the prediction error neurons would make the local learning rule infeasible. On the other hand, to better understand the model, we conducted several detailed experiments, particularly for the first task. Figures 3c, 4b–e, and 6b–d present some of these results.
>
> **Q1.** hEBM demonstrates advantages over the forward model when learning discontinuous functions, as shown in ref [1]. Meanwhile, CANN provides an excellent representation for the action transfer task and enables efficient temporal compression.
>
> **Q2.** In Table 1, we present a comparison with TransDreamer, and in Table 2, we compare with tPCN. These are recent works published in 2022 and 2023 respectively.
>
> **Q3.** The impact of the number of layers on prediction accuracy is illustrated in Figures 3c, 4b–e, 6b–c, and 6f.
>
> ref:
>
> [1] Florence, Pete, et al. "Implicit behavioral cloning." *Conference on Robot Learning*. PMLR, 2022.

---

### Meta-Review · Area_Chair_i3Jn · 2024-12-21

**Metareview:**

The submission posted a solution an interesting topic on the neural mechanism to perform prediction of future observations given current state and memory. Authors proposed an energy based model to preform prediction, inference, and learning. A hierarchical and continuous attractor network is used to achieve these goals. Experimental results show the model can faithfully generate future observations for the environments the model is trained on and for unseen environments.

While the overall setup and experimentation in the paper seems interesting, three reviewers were in favor of rejection due to issues concerning presentation of the paper (citations, uncertainty bars, major contribution) as well as experimentation (reviewer 9CDg)

**Additional Comments On Reviewer Discussion:**

Reviewers and authors had a discussion but reviewers remained unconvinced about the merits of the paper

---

### Decision · Program_Chairs · 2025-01-22

Reject